# Correlation between Irradiation Treatment and Metabolite Changes in *Bactrocera dorsalis* (Diptera: Tephritidae) Larvae Using Solid-Phase Microextraction (SPME) Coupled with Gas Chromatography-Mass Spectrometry (GC-MS)

**DOI:** 10.3390/molecules27144641

**Published:** 2022-07-20

**Authors:** Changyao Shan, Baishu Li, Li Li, Beibei Li, YongLin Ren, Tao Liu

**Affiliations:** 1Institute of Equipment Technology, Chinese Academy of Inspection and Quarantine, No. A3, Gaobeidianbeilu, Chaoyang District, Beijing 100123, China; 34424808@student.murdoch.edu.au (C.S.); libaishu@163.com (B.L.); lili@caiq.org.cn (L.L.); 2College of Science, Health, Engineering and Education, Murdoch University, Perth 6150, Australia; beibei.li@murdoch.edu.au

**Keywords:** *Bactrocera dorsalis* (Diptera: Tephritidae), irradiation treatment, SPME technology, metabolites, metabolic pathway

## Abstract

The metabolites produced by the larvae of *Bactrocera dorsalis* (Diptera: Tephritidae) exposed to different doses of irradiation were analyzed using solid phase microextraction (SPME) and gas chromatography-mass spectrometry (GC-MS), and a metabonomic analysis method of irradiated insects based on GC-MS was established. The analysis revealed 67 peaks, of which 23 peaks were identified. The metabolites produced by larvae treated with different irradiation doses were compared by multivariate statistical analysis, and eight differential metabolites were selected. Irradiation seriously influenced the fatty acid metabolic pathway in larvae. Using the R platform combined with the method of multivariate statistical analysis, changes to metabolite production under four irradiation doses given to *B. dorsalis* larvae were described. Differential metabolites of *B. dorsalis* larvae carried chemical signatures that indicated irradiation dose, and this method is expected to provide a reference for the detection of irradiated insects.

## 1. Introduction

*Bactrocera dorsalis* (Hendel), called oriental fruit fly, is one of the world’s most damaging agricultural pests. Because they have a wide range of hosts, primarily fruits, and are highly invasive. The fruit fly was initially recorded in Taiwan in 1912, and it has since widely distributed throughout the Asia-Pacific region and much of sub-Saharan Africa [1,2]. Through its larvae, which feed on fruit, the species can cause massive direct crop losses to agricultural production. Simultaneously, significant indirect loss is caused by quarantine restrictions on potentially infested fruits [3]. Therefore, *B. dorsalis* is known as a quarantine pest, and it is critical to be aware of quarantine requirements throughout the world.

Irradiation using high-energy rays causes sterility in insects. Irradiation can be generated by gamma rays, high-energy X-rays or high-energy electron beams released by cobalt as the energy source [1,4]. The dose range is generally 100–300 Gy, which is clean, fast and suitable for refrigerated fruits. Irradiation is a proven phytosanitary treatment method. Irradiation of fruit flies in the quarantine environment has been successfully used for many years [5,6,7]. The International Plant Protection Convention Organization (IPPC) has issued a variety of international standards for irradiation treatment doses of fruit flies and is considering 150 Gy as a general dose for fruit flies [8].

Irradiation treatment of insects can prevent their successful reproduction. A difference from other methods of controlling insects is that irradiated insects remain alive, though sterile. Live insects discovered in quarantine situations are assumed to be fertile, and large costs may be required to contain them [9,10,11]. Sterile insects are not a threat to biosecurity, whereas fertile insects are. Thus, it is important to know the radiation status of potentially invasive insects found in biosecurity checks [11]. The development of a protocol to quantify radiation dose was the aim of the research described in this paper.

Various approaches were developed with the aim of determining radiation dosage in insects, but they largely remain at the experimental concept stage [12]. One approach is to observe the increase in abundance of insect gut bacteria, another assesses the damage to genes and chromosomes, and another records changes to the structure of sperm and mitochondria [13,14,15,16,17]. Problems with such approaches are large individual differences in histomorphology; therefore, the sensitivity and application of these approaches on a high-throughput level is problematic. There remains a need to identify a robust, sensitive and reproducible approach to measuring irradiation dose in insects.

In recent years, more attention has been paid to the study of volatile compounds using solid phase microextraction (SPME) which is a rapid sample treatment technology [18,19,20,21]. Compared with traditional detection methods, solid phase microextraction coupled with gas chromatography-mass spectrometry (GC-MS) technology provides increased extract purity, reproducibility and sensitivity [18]. By studying the composition of lipids in the stratum corneum and internal tissues of *Tribolium castaneum* (Herbst) and *Rhyzopertha dominica* (Fabricius), Alnajim et al. (2019) found that SPME-GC-MS is an efficient extraction and sensitive analytical method for the determination of non-derivative insect fats in stratum corneum and homogenate tissues [22]. Al-Khshemawee et al. (2018) used GC-MS to analyze the representative metabolites of insects captured using SPME in order to better understand the biological changes in *Ceratitis capitata* (Wiedemann) during mating [23]. Tanaka et al. (2021) used solid phase microextraction technology to show that volatile isopentenols and polysulfides were biomarkers for early the detection of brown rice diseases and insect pests [24]. Lijun Cai et al. (2022) used SPME-GC-MS to analyze and compare the composition of volatile compounds produced by *T. castaneum*, *R. dominica* and *Sitophilus ranarius* (L.) alone and together with wheat, and identified the biomarkers released by three kinds of wheat stored grain pests [13].

As barriers to international trade in primary produce (food, feed, fibers, timber, etc) fall around the world, prevention of the spread of exotic insect pests and their control has become a high priority [1,25,26]. Irradiation treatment technology has a broad application because of its cleanliness and high efficiency [27,28,29], but some insects survive this treatment. There is a need to rapidly determine the impact that irradiation has had on the reproductive fitness of those insects. Existing detection techniques are cumbersome, insensitive and with poor reproducibility, and so there is an urgent need to develop convenient and fast detection techniques [10]. The purpose of this study was to evaluate the irradiation dose to *B. dorsalis* larvae based on SPME-GC-MS as an alternative approach. We discuss its theoretical basis and its potential benefits over existing methods.

## 2. Results and Discussion

### 2.1. Effect of Irradiation Doses on Emergence Rate of B. dorsalis Larvae

As irradiation dose increases from 0 Gy to 60 Gy, the emergence rate of the 3rd instar of *B. dorsalis* larvae sharply declines (Figure 1), a function of disruption to metabolic pathways [30,31]. When the irradiation dose is 0 Gy, the emergence rate of the 3rd instar of *B. dorsalis* larvae is 94.01 ± 1.23%; when the irradiation dose is 30 Gy, the emergence rate of the 3rd instar of *B. dorsalis* larvae is only 7.84 ± 2.33%; when the irradiation dose reaches or exceeds 60 Gy, the 3rd instar of *B. dorsalis* larvae can hardly emerge normally.

### 2.2. Metabolite Expression in Response to Irradiation Dose

In all samples there were 67 peaks, and 23 metabolites were identified from these by GC-MS raw data (Table 1). For the *B. dorsalis* larvae treated with four levels of irradiation compared with untreated controls, the range of different metabolites produced decreased with the increase in irradiation dose. At the same time, the content of specific metabolites varied with the irradiation dose. Compared with the control group, when the irradiation dose reached or exceeded 30 Gy, the content of benzaldehyde, 3-ethyl-, hexanedioic acid, dioctyl ester, sulfurous acid, 2-ethylhexyl hexyl ester, 1,2,4-Benzenetricarboxylic acid and 1,2-dimethyl ester were significantly down-regulated, while the content of tetradecanoic acid increased. When the irradiation dose reached or exceeded 60 Gy, the content of octadecanoic acid suddenly increased, while the content of 2-Hydroxy-gamma-butyrolactone, Dodecanoic acid, Diethyl Phthalate, n-Hexadecanoic acid and Di-n-decylsulfone rapidly decreased. When the irradiation dose reached or exceeded 90 Gy, the amount of n-Decanoic acid, Phthalic acid and cyclobutyl isobutyl ester increased. When the irradiation dose reached 120 Gy, 2-Hexen-1-ol and (*E*)- increased. Pentadecanoic acid, Butyric acid, 2,2-dimethyl- and vinyl ester were detected only in unirradiated larvae, so this is a marker indicating no treatment. Other compounds could stably exist at the irradiation dose of 0 Gy to 120 Gy, and the content of 6-methylmethyl and Oleic acid slightly decreased with the increase in irradiation dose.

### 2.3. Hierarchical Cluster Analysis

The data set was scaled using the heatmap software package in the R software (Version: 4.1.3), and the samples and metabolites were analyzed using two-way cluster analysis [32]. Figure 2 is a hierarchical cluster diagram of relative quantification of metabolites of *B. dorsalis* larvae. The heatmap is divided into five areas: green, yellow, red, grey and blue, indicating that the content of metabolites greatly varies, and the difference between them is obvious. At the top of the graph, the samples of metabolites of *B. dorsalis* larvae treated with different irradiation doses are clustered. The clustering results clearly show two main irradiation dose clusters (cluster 1): 0 Gy and 30 Gy (cluster 1), and 60 Gy, 90 Gy and 120 Gy (cluster 2). For cluster 1, when the irradiation doses were 0 Gy and 30 Gy, the content of Benzaldehyde, 3-ethyl-, Hexanedioic acid, dioctyl ester, Sulfurous acid, 2-ethylhexyl hexyl ester, Diethyl Phthalate, n-Hexadecanoic acid, Di-n-decylsulfone, 2-Hydroxy-gamma-butyrolactone, Dodecanoic acid, n-Decanoic acid, 1,2,4-Benzenetricarboxylic acid, 1,2-dimethyl ester, Pentadecanoic acid, Butyric acid, 2,2-dimethyl- and vinyl ester metabolites produced significantly increased. For cluster 2, the content of metabolites included 1-Pentene, 4,4-dimethyl-, Linoelaidic acid, 1-Tetradecene, Oxalic acid, 2-ethylhexyl hexyl ester, Oleic acid, Supraene, 2-Hexen-1-ol, (*E*)-, Phthalic acid, cyclobutyl isobutyl ester, Octadecanoic acid, 1-Octene, 6-methyl- and Tetradecanoic acid, and these metabolites were significantly upregulated when the irradiation doses were 60 Gy, 90 Gy and 120 Gy. These results are consistent with the results of the GC-MS analysis in Table 1. The above analyses show that different irradiation doses received by the larvae can be inferred by the presence of different metabolites.

### 2.4. Multivariate Analysis of Metabolites in B. dorsalis Larvae Exposed to Different Irradiation Doses

Multivariate statistical analysis can simplify and reduce the dimensionality of high-dimensional and complex data while retaining a large amount of original information [33]. An unsupervised clustering method, PCA, is implemented on the screened GC-MS data. Six principal components are obtained with a cumulative contribution rate of 92.29%, indicating that the fitting degree of the PCA model is high, and the results of multidimensional statistical analysis are reliable. Therefore, the PCA model can be used to analyze the overall differences between treatment groups and the differences between samples within groups. The PCA scores chart (Figure 3A) showed that there were significant differences between larvae treated with different irradiation doses (30 Gy, 60 Gy, 90 Gy and 120 Gy) and the control group (0 Gy). There was significant separation between the treatment group and the control group on the first principal component (Component 1), which could explain 49.36% of the total variance. However, the separation effect between the treatment groups with irradiation dose of 30 Gy and 60 Gy was not significant.

Partial least-squares discriminant analysis (PLS-DA) is a multidimensional statistical analysis method for supervised pattern recognition [34,35]. Compared with PCA, PLS-DA not only reduces the dimension, but also combines the regression model to make a discriminant analysis of the regression results with a certain discriminant threshold, which is helpful to identify differences in compounds between groups. To maximize separation between groups, PLS-DA was carried out on the basis of the above GC-MS data, which better illustrated the differences in metabolites produced by the larvae upon exposure to different irradiation doses. Figure 3B shows the classification pattern using the partial least square model, and the variance explains 60.63% and 13.51% of Component 1 and Component 2, respectively. Using the different compounds produced by the larvae after different irradiation doses, all larvae can be effectively isolated. Larvae treated with 60 Gy, 90 Gy and 120 Gy gathered on the left side of the score map, while those treated with 0 Gy and 30 Gy gathered on the right side of the score map. Thus, the metabolite analysis using PCA and PLS-DA completely correspond to that using hierarchical cluster analysis (HCA).

A variable importance projection (VIP) score was constructed, in which VIP > 1 represents important distinguishing compounds to further identify the key compounds that are differentially present in larva treated with different irradiation doses [36,37]. The partial least square VIP scores of 23 metabolites are shown in Figure 3C. Table 2 shows irradiation doses and eight metabolites with significant changes determined using partial least square VIP score and t-test p value, which are n-Hexadecanoic acid, Dodecanoic acid, Octadecanoic acid, Tetradecanoic acid, 1,2,4-Benzenetricarboxylic acid, 1,2-dimethyl ester, Butyric acid, 2,2-dimethyl-, vinyl ester, Phthalic acid, cyclobutyl isobutyl ester, 2-Hexen-1-ol and (*E*)-. These compounds can be used as biomarkers to identify the irradiation dose of *B. dorsalis* larvae during quarantine treatment. When the irradiation dose reached 30 Gy and 120 Gy, the contents of Tetradecanoic acid, Octadecanoic acid, Phthalic acid, cyclobutyl isobutyl ester, 2-Hexen-1-ol, (*E*)- significantly increased, but the changes in Phthalic acid, cyclobutyl isobutyl ester, 2-Hexen-1-ol and (*E*)- contributed relatively little to the discrimination in irradiation dose. When irradiation dose exceeded 0 Gy, 30 Gy and 60 Gy, the contents of Butyric acid, 2,2-dimethyl-, vinyl ester, 1,2,4-Benzenetricarboxylic acid, 1,2-dimethyl ester, Dodecanoic acid and n-Hexadecanoic acid decreased, enabling the prediction of irradiation dose. These compounds significantly changed with the change in irradiation dose, indicating that irradiation treatment disturbed the normal metabolism of *B. dorsalis* larvae. Therefore, we can use the information contained in these differential metabolites to provide a reference for the detection of irradiation dose.

### 2.5. Preliminary Pathway Analysis of Differential Metabolites

The metabolic pathways in insects is regulated by a variety of compounds and reactions. Seven metabolic pathways were examined using the pathway database at the Kyoto Encyclopedia of Genes and Genomes (KEGG) [38,39]. The correlation between differential metabolites and three key metabolic pathways of fatty acid biosynthesis, fatty acid elongation and fatty acid degradation were screened, as shown in Table 3. The metabolic pathways of differential metabolites after treatment with different irradiation doses were analyzed using enrichment analysis and KEGG metabolic pathway retrieval. In Figure 4A, the number of bubbles represents the number of metabolic pathways, the color of the bubbles represents the degree of enrichment and the size of the bubbles represents the total number of metabolites contained in the metabolic pathway.

Metabolic pathway analysis revealed that irradiation affected fatty acid metabolism, indicating that there was a high correlation between the difference in fatty acid content and irradiation dose, but the mechanism for this was unclear [40]. According to previous research, fatty acids are basic substances present during insect embryonic development, metamorphosis and other life activities involved in growth, development and reproduction. [41,42]. Development of *B. dorsalis* larvae can be indirectly regulated by irradiation dose through the change in fatty acid content in the body, which is consistent with the results shown in Figure 1.

Two differential metabolites found in the target metabolic pathway were n-Hexadecanoic acid and Tetradecanoic acid (Figure 4B). The metabolites n-Hexadecanoic acid and Tetradecanoic acid are involved in the metabolism of fatty acid biosynthesis; n-Hexadecanoic acid is a common key metabolite in the two metabolic pathways of fatty acid elongation and fatty acid degradation. With the increase in irradiation dose, the content of n-Hexadecanoic acid released decreased, while the content of Tetradecanoic acid showed the opposite trend. This may be due to *n*-Hexadecanoic acid being partly converted to Tetradecanoic acid and ethyl easter at a high dose of irradiation treatment. Considering the pathways and mechanisms of fatty acid synthesis in insects, this result may be caused by the enhancement of the activity of key enzymes such as Acetyl CoA carboxylase or fatty acid synthase, leading to a certain degree of lipid accumulation [43,44,45,46]. Moreover, n-Hexadecanoic acid and Tetradecanoic acid were found to be responsible for the insect growth inhibitory and insecticide activity [47,48,49,50]. Therefore, irradiation treatment can lead to fatty profile changes and was found to be lethal to treated insects, but the mechanism was more complicated than we expected.

## 3. Materials and Methods

### 3.1. Insect Culture

Insects used in this study were collected from a mango orchard in Guangxi Zhuang Autonomous Region, China. They were reared in a laboratory at the Chinese Academy of Inspection and Quarantine. Late 3rd instar *B. dorsalis* larvae that emerged from mango fruit were transferred to moist sterile sand for pupariation. The pupae were placed in rearing cages (40 × 40 × 50 cm). Adults were fed with orange slices and a solid mixture of sucrose and hydrolyzed yeast (3:1) [51]. Eggs were collected from two weeks after adults had emerged from the puparium and mated. The adult females laid eggs through the sides of the cage cloths and the eggs fall into a distilled water (26 ± 1 °C) collector. Between 20–30 mL of egg suspension containing more than 7000 eggs were produced in 8 to 12 h. The larvae were reared on the artificial diet described by Vargas et al. (1984) [52]. At all stages, the larvae were reared at 25 ± 2 °C and relative humidity 70 ± 5% with a photoperiod of 12:12 (D: L) h.

### 3.2. Irradiation Treatment

All irradiation treatments were performed in an RS-2000 ProX irradiator (Rad SourceTechnologies, Inc., Coral Springs, FL, USA). The operation parameters were 220 KV and 17.6 mA [53]. A plastic box containing larvae was placed in the irradiation room with given irradiation doses of 0 Gy, 30 Gy, 60 Gy, 90 Gy and 120 Gy. The dose rate was 5.0 Gy/min for irradiation exposure time of 6 min, 12 min, 18 min and 24 min. Three or five plastic boxes (as replicates) were placed in the irradiation chamber and were irradiated at the same time. After irradiation, larvae were placed in a constant temperature incubator for 1 h before metabolites were extracted using SPME.

### 3.3. Solid Phase Microextraction (SPME) Procedure and Sampling Setup

The cuticular compounds profile of the late 3rd instar of *B. dorsalis* larvae were obtained by gently rubbing the body surface with a 50/30 um DVB/CAR/PDMS fiber solid phase microextraction sampler. For sampling the surface compounds, a new tool was designed which allows fixing larvae and adjusting the fiber position (Figure 5). Six slide glasses were placed as shown in Figure 5 using a hot-melt adhesive to hold one side of the slide glasses and then adjusting the slide glass on the other side to form an angle between these two slides [54]. With this structure, the larvae were fixed onto this tool and the SPME syringe was pushed into the slit. The fiber was slowly moved toward the larvae until the absorbent portion of the fibers was fully attached to the entire body segments of the larvae. Then, the slide was moved back and forth so that the body surface of the larvae constantly rubbed against the absorbent portion of fiber for 5 min.

### 3.4. Gas Chromatography-Mass Spectrometry (GC-MS) Conditions

Agilent 8890 gas chromatograph (GC) was used with an HP-5MS capillary column (30 m × 0.25 mm, 0.25 μm; Agilent J&W Scientific) and a 5977B mass selective detector (MSD). The carrier gas used was 99.999% purified helium with a constant flow rate of 1 mL/min. The GC conditions were: injection temperature of 270 °C and column temperature program of 60 °C for 5 min, which was then increased to 180 °C at the rate of 5 °C/min, and was finally increased to 280 °C at the rate of 10 °C/min, and maintained for 5 min. The MS parameters were: the transmission line temperature of the ion source was 280 °C and the quadrupole temperature was 150 °C. The information was collected using the full scanning mode of the mass spectrometry, the mass scanning range of the mass spectrometry was 30,500 atomic mass units (Amu) and the solvent delay time was 4.5 min. The total running time was 49 min.

### 3.5. Statistical Analysis

The GC-MS data were preliminarily identified using Aglient MasterHunter Qualitative Analysis 10.0 and recorded and sorted in Microsoft Excel. Most metabolites were further identified using the National Institute of Standards and Technology (NIST) and Wiley Registry of mass spectral data, as well as the retention index provided by the compound database of NIST Chemistry Web Book [55].

The XCMS package in RStudio (Version: 2021.9.0) was used to extract and analyze the feature data of GC-MS data [39]. The edited data matrix was imported into RStudio, and the unsupervised PCA and supervised PLS-DA were analyzed in R (Version: 4.1.3) prcomp package and mixOmics package. The differential metabolites were screened according to VIP-value and Student *t*-test, and the significant differences among the experimental groups were analyzed [35,56]. The selected differential metabolites were annotated in KEGG, and all the pathways of differential metabolites mapping were retrieved [57]. Then, the pathways of differential metabolites were further screened using enrichment analysis in order to find the key pathways with the highest correlation with differential metabolites.

## 4. Conclusions

In this experiment, a simple tool was used to fix *B. dorsalis* larvae, and SPME was used to extract metabolites from larvae exposed to different doses of irradiation treatment. Metabolites of different treatment groups were identified using GC-MS. By comparing the metabolites produced after five different irradiation doses, it was found that there were eight differential metabolites produced, of which four were fatty acids. The differential presence of metabolites occurred with different irradiation doses, and it was feasible to infer the irradiation dose of larvae using the difference in metabolites. This method is feasible to separate and identify the metabolites from *B. dorsalis* larvae in the process of quarantine treatment.

In summary, irradiation dose influences the spectrum of metabolites present in *B. dorsalis* larvae. Metabolic pathway analysis showed that different irradiation doses significantly changed fatty acid-related metabolic pathways in the species tested. Specifically, it was found for the first time that opposite trends occurred to n-Hexadecanoic acid and Tetradecanoic acid; that is, when *n*-Hexadecanoic acid decreased, Tetradecanoic acid increased. These findings provide a basis for understanding the molecular mechanism of irradiation damage in insects, and a basis for developing biomarkers corresponding to different irradiation doses.

## Figures and Tables

**Figure 1 molecules-27-04641-f001:**
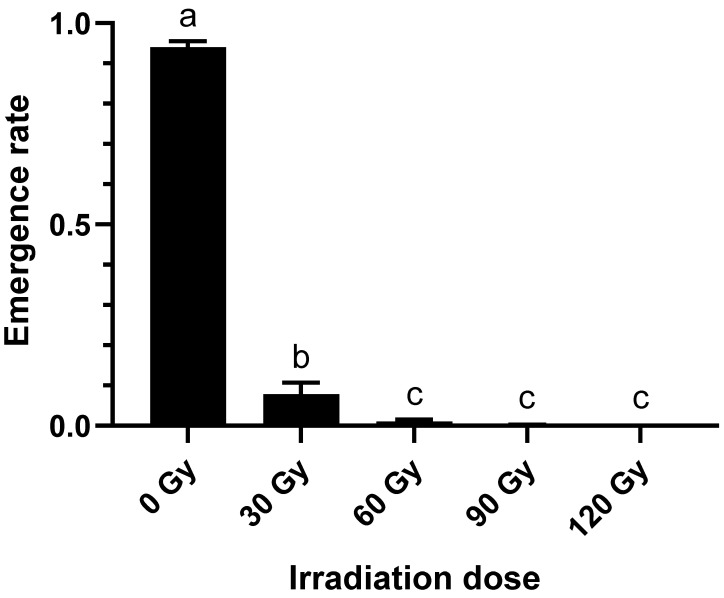
Emergence rates of the 3rd instar of *B. dorsalis* larvae in response to irradiation dosage (Different lower case letters indicate that emergence rates has significant differences between irradiation dosages at *p* < 0.05).

**Figure 2 molecules-27-04641-f002:**
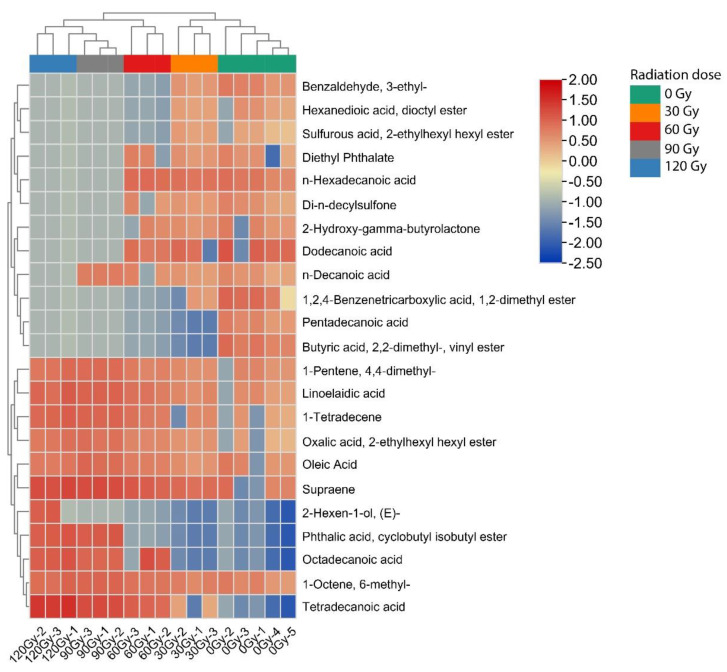
Clustering heatmap of metabolites in *B. dorsalis* larvae exposed to different doses of irradiation treatment.

**Figure 3 molecules-27-04641-f003:**
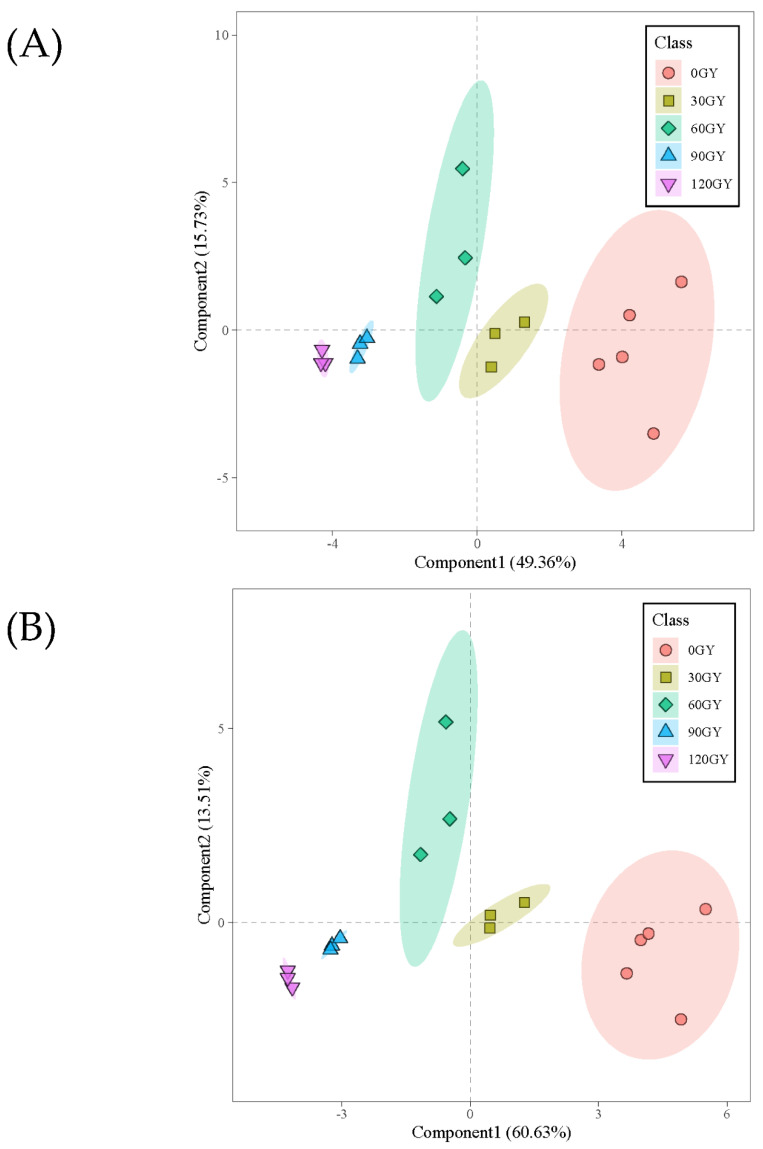
Principal component analysis (PCA) score plot (**A**), partial least squares-discriminant analysis (PLS-DA) loading plot (**B**), and variable importance projection (VIP) scores plot (**C**) of all metabolites in *B. dorsalis* larvae exposed to different doses of irradiation treatment at 0 Gy, 30 Gy, 60 Gy, 90 Gy and 120 Gy.

**Figure 4 molecules-27-04641-f004:**
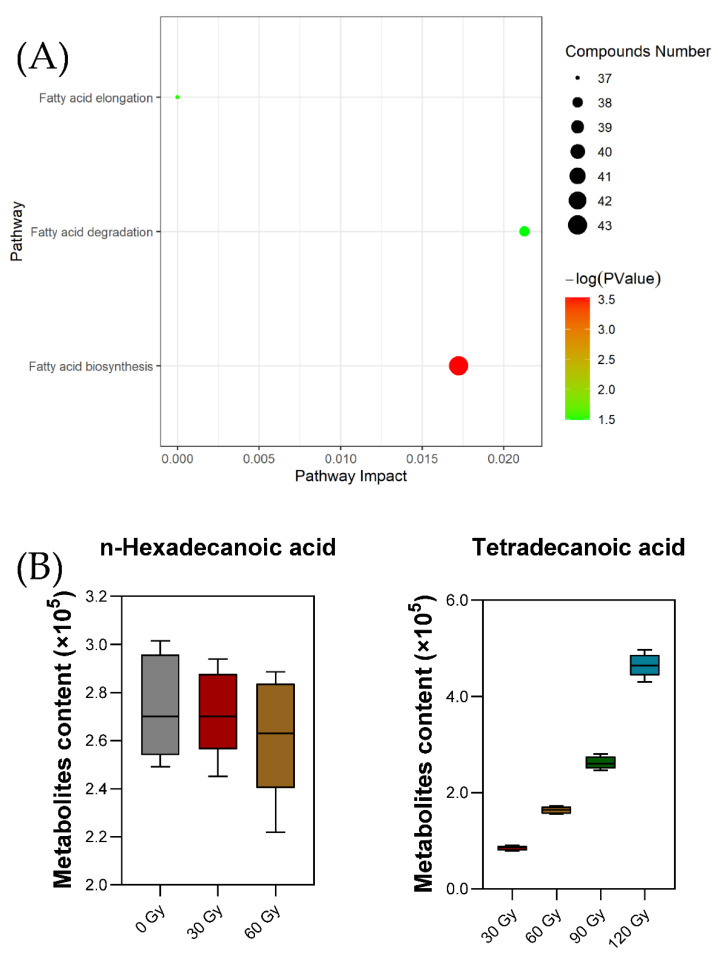
Effects of hit metabolites on metabolic pathways in *B. dorsalis* larvae (**A**). Content changes in metabolites in key metabolic pathways (**B**).

**Figure 5 molecules-27-04641-f005:**
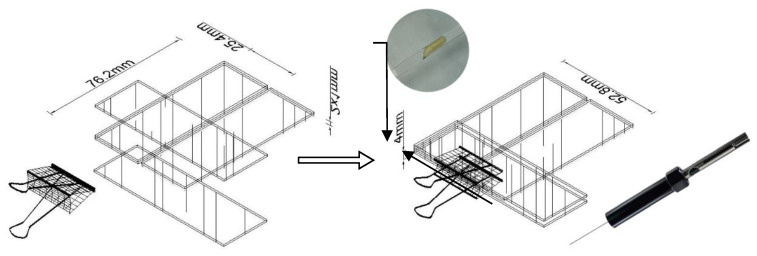
Experimental setup for SPME sampling of *B. dorsalis* larvae specimens. The larvae are fixed in a slit assembled by slide glasses and contacted with an SPME fiber.

**Table 1 molecules-27-04641-t001:** Metabolites produced by *B. dorsalis* larvae treated with different irradiation doses.

Metabolites	Retention Time	Retention Index	GC Response (10^5^) ± SD					CAS Number
			0 Gy	30 Gy	60 Gy	90 Gy	120 Gy	
2-Hydroxy-gamma-butyrolactone	8.404	997.26	1.2 ± 0.06	1.12 ± 0.13	1.06 ± 0.14	N.D.	N.D.	19444-84-9
1-Pentene, 4,4-dimethyl-	9.825	1037.86	1.41 ± 0.12	1.25 ± 0.15	1.14 ± 0.13	1.26 ± 0.16	1.01 ± 0.10	762-62-9
Benzaldehyde, 3-ethyl-	14.305	1061.02	1.36 ± 0.17	0.91 ± 0.11	N.D.	N.D.	N.D.	34246-54-3
2-Hexen-1-ol, (*E*)-	15.58	1102.61	N.D.	N.D.	N.D.	N.D.	2.11 ± 0.09	928-95-0
*n*-Decanoic acid	20.686	1274.26	1.14 ± 0.16	1.09 ± 0.13	0.89 ± 0.10	0.67 ± 0.02	N.D.	334-48-5
1-Octene, 6-methyl-	21.288	1295.34	1.55 ± 0.16	1.54 ± 0.12	1.29 ± 0.26	1.16 ± 0.12	1.05 ± 0.14	13151-10-5
1-Tetradecene	23.28	1369.70	1.7 ± 0.10	1.62 ± 0.01	1.27 ± 0.12	1.23 ± 0.08	1.39 ± 0.16	1120-36-1
Dodecanoic acid	24.966	1435.12	3.72 ± 0.25	3.31 ± 0.17	1.13 ± 0.13	N.D.	N.D.	143-07-7
Diethyl Phthalate	25.772	1467.36	1.32 ± 0.12	1.06 ± 0.08	1.04 ± 0.06	N.D.	N.D.	84-66-2
Tetradecanoic acid	29.365	1739.12	N.D.	0.85 ± 0.10	1.64 ± 0.10	2.62 ± 0.23	4.6 ± 0.43	544-63-8
Pentadecanoic acid	31.94	1817.30	1.45 ± 0.13	N.D.	N.D.	N.D.	N.D.	1002-84-2
Phthalic acid, cyclobutyl isobutyl ester	35.306	1947.82	N.D.	N.D.	N.D.	1.66 ± 0.17	1.61 ± 0.22	1000314-91-1
*n*-Hexadecanoic acid	35.23	2058.07	2.73 ± 0.19	2.72 ± 0.12	2.63 ± 0.28	N.D.	N.D.	57-10-3
Linoelaidic acid	35.969	2135.05	1.54 ± 0.21	1.53 ± 0.19	1.14 ± 0.09	1.28 ± 0.11	1.19 ± 0.03	506-21-8
Oleic acid	38.664	2181.34	1.68 ± 0.23	1.35 ± 0.18	1.09 ± 0.1	0.98 ± 0.09	0.89 ± 0.03	112-80-1
Octadecanoic acid	39.014	2187.36	N.D.	N.D.	1.08 ± 0.04	1.23 ± 0.01	1.56 ± 0.21	57-11-4
Oxalic acid, 2-ethylhexyl hexyl ester	40.585	2221.02	1.22 ± 0.16	1.08 ± 0.10	0.85 ± 0.21	0.83 ± 0.07	0.91 ± 0.08	1000309-38-9
Hexanedioic acid, dioctyl ester	41.759	2350.94	1.27 ± 0.13	0.79 ± 0.11	N.D.	N.D.	N.D.	123-79-5
Di-*n*-decylsulfone	42.046	2439.44	1.57 ± 0.17	1.16 ± 0.14	0.96 ± 0.07	N.D.	N.D.	111530-37-1
Sulfurous acid, 2-ethylhexyl hexyl ester	43.309	2561.14	1.32 ± 0.17	0.9 ± 0.04	N.D.	N.D.	N.D.	1000309-20-2
1,2,4-Benzenetricarboxylic acid, 1,2-dimethyl ester	44.34	2678.85	3.26 ± 0.21	1.04 ± 0.11	N.D.	N.D.	N.D.	54699-35-3
Butyric acid, 2,2-dimethyl-, vinyl ester	45.219	2793.95	2.13 ± 0.14	N.D.	N.D.	N.D.	N.D.	13170-00-8
Supraene	45.568	2867.19	2.69 ± 0.29	2.82 ± 0.01	2.04 ± 0.25	2.65 ± 0.26	2.39 ± 0.31	7683-64-9

SD: standard deviation. N.D.: metabolite not detected.

**Table 2 molecules-27-04641-t002:** Significantly changed metabolites in *B. dorsalis* larvae exposed to different doses of irradiation treatment.

Metabolites	VIP Scores	*p* Value	FDR	Class
*n*-Hexadecanoic acid	1.505	0.049	0.069	Acid
Dodecanoic acid	1.421	0.003	0.008	
Octadecanoic acid	1.371	0.001	0.006	
Tetradecanoic acid	1.355	0.005	0.012	
1,2,4-Benzenetricarboxylic acid, 1,2-dimethyl ester	1.120	0.003	0.008	Ester
Butyric acid, 2,2-dimethyl-, vinyl ester	1.098	0.003	0.008	
Phthalic acid, cyclobutyl isobutyl ester	1.056	0.000	0.000	
2-Hexen-1-ol, (*E*)-	1.009	0.017	0.029	Alcohol

**Table 3 molecules-27-04641-t003:** The key metabolic pathways of differential metabolites in *B. dorsalis* larvae.

Pathway Name	Total	Expected	Hits	Raw *p*	Impact	Hit Metabolites
Fatty acid biosynthesis	43	0.28289	2	0.029454	0.01724	C00249: *n*-Hexadecanoic acidC06424: Tetradecanoic acid
Fatty acid elongation	37	0.24342	1	0.22001	0	C00249: *n*-Hexadecanoic acid
Fatty acid degradation	38	0.25	1	0.22532	0.02128	C00249: *n*-Hexadecanoic acid

## Data Availability

All data are contained within the article.

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
