# Peer review of "Correlation between Irradiation Treatment and Metabolite Changes in Bactrocera dorsalis (Diptera: Tephritidae) Larvae Using Solid-Phase Microextraction (SPME) Coupled with Gas Chromatography-Mass Spectrometry (GC-MS)"

_molecules, 2022, doi:10.3390/molecules27144641_

Round 1
Reviewer 1 Report
The authors have revised the manuscript and improved the quality of the article. However, I still have some problems with the biochemical/physological interpretation of the analytical data, i.e. mainly fatty acid profiles as a function of radiation treatment. I understand that a final conclusion is not possible based on the available data, however, the authors should at least briefly discuss their data taking into account basic pathways and mechanism in the formation of saturated resp. unsaturated fatty acids in insects.
Reviewer 2 Report
Thanks for addressing my comments! COngrats on the publication!
Author Response
It is certainly good comments and useful suggestion. Thanks for your encouragement!
This manuscript is a resubmission of an earlier submission. The following is a list of the peer review reports and author responses from that submission.
Round 1
Reviewer 1 Report
All comments are given in the manuscript. The authors should make an attempt to identify one or two metabolites that can surely go as biomarkers with the available data.
regds

Reviewer 2 Report
Shan et al. provide an interesting study about changes in the metabolite profiles obtained from larvae of the oriental fruit fly Bactrocera dorsalis when exposed to different doses of irradiation. Here, the authors focused on 8 key metabolites from solid phase extraction and subsequent GC-MS analysis. The data were subjected to multivariate statistical analysis which revealed that the metabolism of fatty acids was significantly changed in a radiation dependent manner. The analytical work has been done carefully, although the limited number of metabolites which can be taken for the interpretation of the biological effects is disappointing. Another weakness of the manuscript is the limited interpretation of the biochemical pathways being affected. Here, a more elaborated analysis and correlation of the metabolite profiles with the underlying biochemistry is appropriate. On this basis, the study seems to be premature and descriptive. More metabolites (additional extraction protocols) should be included for assigning the effects with a more detailed interpretation of the biochemistry. Therefore, I am afraid that the study should not be published in the present state, but the authors should be recommended to resubmit an extended version of their work.
Reviewer 3 Report
Shan et al. reported a comparative metabolomics analysis to study the effect of irradiation of a fruit fly using SPME and GC-MS. In general, the article is well written with sufficient background in the introduction which helps the reader to get the importance of the performed study. So, I thank the authors for this. I recommend accepting this article once my following comments are addressed by the author.
1. Page 2 line 38: Sentence starts with a lower-case letter. Please correct this.
2. Can the authors clarify what is the identification mode mean in table 1?
3. Page 6 line 170: A comma is needed between 60 Gy and 90 Gy.
4. What is HCA on page 6 line 173?
5. What was the irradiation exposure time? This should be mentioned in 3.2
6. After SPME, were the larvae alive?